# The prevalence of suicidal behavior and its associated factors among wives with polygamy marriage living in Gedeo zone, southern Ethiopia, 2020

Chalachew Kassaw, Seid Shumye \* 

Department of Psychiatry, College of Health Science, Dilla University, Dilla, Ethiopia

\* seidshumye22@gmail.com

## Abstract

### Introduction

Polygamy is a trend of marriage characterized by having two and more wives or husbands at the same time. In low and middle-income countries including Ethiopia, polygamy has a significant negative effect on the social, economic, physical, and mental well-being of women. Therefore, this study aimed to assess the prevalence and associated factors of suicidal behavior among wives with polygamy marriage living in the Gedeo zone, Southern Ethiopia, 2020.

### Methods

A community-based study employing cross-sectional design and systematic sampling technique was used to select wives with polygamy marriage who are residents of Gedeo Zone from November to December 2020. The World Health Organization Suicidal Behavior Questions (SBQ-5) was adapted to explore the outcome variable. The Logistic regression at 95% CI, p<0.05 was used to identify factors associated with suicidal behavior.

### Results

This study enrolled 423 respondents. The study revealed that, the overall prevalence of suicidal behavior was 157(37%). Illiteracy, being a wife of a husband with three and more other wives, current history of depression, intimate partner violence, and poor social support were significantly associated with suicidal behavior at 95% confidence interval, p < 0.05.

### Conclusion

This study found that one-third of the respondents had suicidal behavior. Different significant socio-demographic and psychosocial variables were identified. Thus, due attention should be given to minimize the practice and its effect on the mental wellbeing of a mother and their children.

**Data Availability Statement:** Data cannot be shared because it is potentially replicable by others and contain potentially identifying information. The data restriction was imposed by the Institutional

Review Board and for further information you can contact one of the committee via this email address:gizsisay@gmail.com and you can access the data upon request.

**Funding:** The author(s) received no specific funding for this work.

**Competing interests:** The authors have declared that no competing interests exist.

**Abbreviations:** AOR, Adjusted odd ratio; CI, Confidence interval; OSLO, 3- Oslo social support scale; PHQ, Patient health questioner; SBQ-R, Suicidal behavioral revised questioner; SD, Standard deviation.

# Introduction

Suicidal behavior is a pattern of thinking or predispositions that situated an individual at risk of committing suicide. Suicidal behavior has three categories; suicide ideation, suicide plan or intent, and suicide attempts which are the leading cause of harm and end of life [1]. Suicide is one of the complications of a psychiatric disorder and affecting all individuals irrespective of their nation, culture, religion, gender, and class [2]. Currently, Lithuanians rank first for a suicidal rate, which accounts 31/100k [3]. Globally, while over 16,000,000 people attempt suicide, and 800,000 commit a suicide every year, other 3000 individuals also commit suicide in a daily base. The case is more prevalent among youths and women [4].

Polygamy is a state of marriage in which a husband marries more than two wives at a time and remains one of the hotly contested and acceptable practices. It is a misunderstood topic in the traditional and modern society of Africa and Ethiopia. It is acceptable in most cultures and religions, particularly in Muslim [5–7]. It is a very problematic and well-buried practice against freedom, equality, and human rights of a woman [8]. It has also a burden on husbands to care for all their wives and children in terms of social, economic, psychological, and physical needs [9].

Psychological and social factors such as jealousy, intimate partner violence, lack of social support, and economic dependency on their husband were contributing factors for having mental disorders among women with polygamy marriage [10–12]. Several studies have documented that woman with polygamous marriages were reported of having repeated physical and psychological violence from their husbands and thus, children expressed their feelings through pain, hurt, and anger. Moreover, in low and middle-level countries, it has a risk for having sexually transmitted diseases, including HIV/AIDS which also contribute for mental health problems on wives with polygamous marriage, their family and society at all [13, 14]. Psychiatric problems such as mood, anxiety, and psychosis were documented among woman with polygamy marriage in particular among senior wives and their family members [15, 16]. Developed countries such as Canada, USA and Australia were launched human and legal rights to prevent any forms of women abuse and violence happened during marriage [17]. This resulted in reduction of polygamy marriage. However, in Ethiopia polygamy cultural practices accounts 12.1% and its effect on the mental and psychological health of wives is not well-studied [18]. Therefore, this study aimed to assess the prevalence and associated factor of suicidal behavior among wives with polygamy marriage living in Gedeo zone, southern Ethiopia, 2020.

# Methods

## Study design, setting and period

This study was a community-based cross-sectional design conducted among wives with polygamy marriage and lives in Gedeo zone. The data collection period was from November to December 2020. Despite being unlawful and excluded from family and criminal code of Ethiopia, polygamy is quiet practiced due to the cultural and religious (Muslims and pagans) acceptability [19]. In Ethiopia, there are 86 nation and nationalities that have unique attitude and norm towards different social circumstances. polygamy is the most acceptable social norm among Oromo, Somalia, Sidama and Gedeo nations [20]. In Gedeo people, there was ancient ancestor called "Daraso" who practiced polygamy in the first time and starting from that a man with polygamy marriage was considered as rich, famous and acceptable by the community [21]. The zone has a population of 850, 534 and 500,000 of them were females. The fertility rate in this study area was higher as compared to other parts of the country [22]. The area

located in the southern part of the country and 359 KM far from the capital city of Ethiopia, Addis Ababa.

## Study participants

The inclusion criteria were women who are presently in polygamous marriage and age 18 and above years old, while the exclusion criteria were women who presented themselves with acute or severe illnesses and not present during the data collection period.

## Sample size determination

The sample size of the current study was calculated using single population proportion formula,

$$n = (Z\alpha/2)pq/d^2 \text{ where.}$$

Where, n = required sample size, z is reliability coefficient at 95% confidence interval (1.96), p = 0.5 (proportion for unknown prevalence of outcome variable, q (1-p) = 0.05, d (margin of error) = 0.05
= (1.96) (1.96) (0.5) (0.5) / (0.05) (0.05) = 384, 10% non-response rate = 38.4
The total sample size was, 384+ 38.4 = 423

## Sampling technique

This study used a systematic random sampling technique. To get the sampling frame of the sample, we gave coding for each number households practicing polygamy marriage. There were 1692 households who practiced polygamy and living in the Gedeo zone. To calculate $K^{th}$ interval, we used the formula (N/n), where N = Total number of population and n = required sample. $K^{th}$ interval = (1692/423 = 4th) and include all samples by counting every $4^{th}$ interval.

## Study variables

**Dependent/out-come variable.**   Suicidal behavior (Yes/No)
**Independent variables.**   *Socio-demographic variables*. Age, educational status, employment, residence, husband work, wife's house and religion.
Marriage and psycho-social variables: Duration of marriage, number of wives at a time, distance of the two wife's house, family size, social support, and duration of marriage.

## Data collection and instruments

The data was collected by ten health extension staffs who are working in the zonal health center. The data collection method was structured interview technique. The first part of the questionnaire was about the socio-demographic characteristics of the respondents. The second part of the questioner was suicidal behavioral revised questionnaire (SBQ-R) used to assess suicidal behavior of the respondents related to polygamy marriage, and a score $\geq$ of 7/18 considered as suicidal behavior [23]. The tool has 93% sensitivity and 95% specificity. The third part of a questionnaire was Intimate Partner Violence (IPV) used to measure intimate partner violence and score > 10/25 classified as intimate partner violence [24]. The fourth part of the questionnaire was the Patient Health Questionnaire (PHQ)−9 tool used to assess depression, and a score > 10/21 said to have depression [25]. The final part of the questionnaire was the Oslo social support scale (Oslo -3) used to measure social support levels from participants friends, neighbors and bloody families. This measurement has been used in both clinical and population-based studies with good internal consistency. In the present study, the tool has

Cronbach alpha of 0.87. The total score ranging from 3 to 14 and classified as poor (3–8), moderate (9–11), and strong social support (12–14) [26].

### Data quality control

This study used reliable questionnaire which has been translated to Amharic and Gedeoffa local languages and showed good internal consistency. The issues written on the questioner were easily understandable by respondents and finished on planned time duration. The validity of the questioner was checked during pretest on 5% of respondents before two weeks of the actual data collection period. The data collected from each respondent has been putted keeping confidentiality/security. The respondents have informed about the purpose and objective of the study before the actual data collection. The collected data checked at daily basis for its completeness.

### Data management and analysis

Data was entered into the Epi-Data version 3.4 software package and exported to the Statistical Package for Social Sciences (SPSS) version 22. Descriptive statistics; frequency, percentage, mean and standard deviation were used to describe the socio-demographic characteristics of respondents. A logistic regression analysis at $p \leq 0.05$ of the 95% CI was used to interpret the association with the independent variables.

### Ethical consideration

Ethical clearance was obtained from the ethical review board of Gedeo zonal women and child health beauro. Each participant has given written informed consent to participate in the current study. The study ensured issues of voluntary participation and confidentiality throughout the data collection period.

## Result

### Socio-demographic results of respondents

The response rate for this study was 100%. The mean (SD) age of respondents was 30 (±6) years old and more than half of the respondents were living in rural areas. Nearly two-thirds of them have no formal education. More than two third, 329(77.8%) of them were protestant religion followers. Regarding to wives work, 250(59%) of them were house wife. Among 423 respondents, 182(43%) of them had poor social support. Of them, 212(50%) of the respondents were living with their husband for at least 6 years (Table 2).

### Psycho-social variables

According to this study result, 237 (56%) of respondents had depression and 186 (44%) intimate partner violence (Fig 1).

### Prevalence of suicidal behavior

Among 423 respondents, 157(37%) of them scored 7/18 for revised suicidal behavior scale and considered as having suicidal behavior (Table 1).

### Factors associated with suicidal behavior

During multivariable logistic regression analysis, at 95% CI (p< 0.05), the independent variables such as educational status, number of wives, history of depression, intimate partner violence, and poor social support were associated with the outcome variable (Table 2).

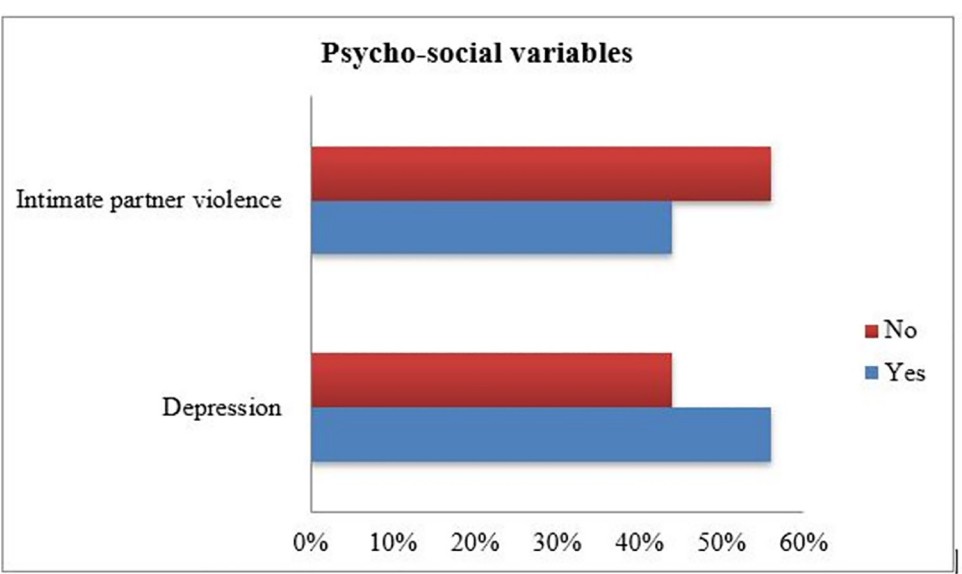

**Fig 1. Descriptive psycho-social results of respondents Dec, 2020 (N = 423).**

**Table 1. Suicidal behavior items response of respondents Dec, 2020 (N = 423).**

| Serial no. | Question items | Response | Frequency | Percentage |
|---|---|---|---|---|
| 1. | Life time suicidal ideation, intent, and/or attempts | Never | 199 | 47% |
| | | Suicidal ideation | 114 | 27% |
| | | Suicide plan | 42 | 10% |
| | | Suicide attempt | 68 | 16% |
| | | Overall suicidal ideation and attempt | 224 | 52.9% |
| 2. | Frequency of suicidal ideation in the past one year | Never | 218 | 51.5% |
| | | Rarely (1 time) | 112 | 26.4% |
| | | Sometimes (2 times) | 51 | 12.1% |
| | | Often (3–4 times) | 25 | 6% |
| | | Very often (5 or more times) | 17 | 4% |
| | | One year all suicidal ideation | 205 | 48.4% |
| 3. | Threat of suicidal attempt | Never | 322 | 76% |
| | | Once | 59 | 14% |
| | | Two and more | 42 | 10% |
| 4. | Likelihood of suicide behavior in the future | Never | 164 | 38.7% |
| | | No chance at all | 47 | 11.2% |
| | | Rather unlikely | 41 | 9.7% |
| | | Unlikely | 18 | 4.3% |
| | | Likely | 72 | 17% |
| | | Rather likely | 51 | 12% |
| | | Very likely | 30 | 7.1% |
| 5. | Overall prevalence of suicidal behavior | Yes | 157 | 37% |
| | | No | 266 | 63% |

**Table 2. Variables associated with suicidal behavior, Dec, 2020 (N = 423).**

| Variables | Category variables | | Suicidal behaviour | | $X^2$ | Adjusted odds ratio (AOR), 95% CI | P-value |
|---|---|---|---|---|---|---|---|
| | | | Yes | No | | | |
| **Age** | Below 30 | 135 | 94 | 41 | 3.1205 | 1.48(0.96, 2.29) | 0.43 |
| | Above 30 | 288 | 175 | 113 | | 1 | |
| **Educational status** | Non formal(illiterate) | 309 | 211 | 98 | 34.7904 | 3.83 (2.44–6.02) | 0.001* |
| | Primary school (grade 1–8) | 114 | 41 | 73 | | 1 | |
| **Residence** | Urban | 42 | 19 | 23 | 3.22 | 0.53(0.28,1.01) | 0.13 |
| | Rural | 381 | 232 | 149 | | 1 | |
| **Religion** | Protestant | 329(77.7%) | 172 | 157 | 1.6788 | 1 | |
| | Orthodox | 64(15.1%) | 29 | 35 | | 1.32(0.77–2.26) | |
| | Muslim | 30(7.09) | 14 | 16 | | 1.252(0.59–2.64) | |
| **Husbands work** | Farmer | 313 | 140 | 173 | 0.83 | 1 | |
| | Merchant | 110 | 43 | 67 | | 0.79(0.51,1.24) | 0.24 |
| **Wives work** | House wife | 250 | 134 | 116 | 5.71 | 1 | |
| | Farmer | 89 | 39 | 50 | | 0.68(0.41,1.10) | 0.10 |
| | Merchant | 84 | 52 | 32 | | 1.41(0.85, 2.33) | 0.37 |
| **Number of wives at a time** | Two | 300 | 132 | 168 | 13.4 | 1 | |
| | Three | 123 | 79 | 44 | | 2.29(1.48,3.53) | 0.01* |
| **Distance of the two wives house** | < 20 km | 148 | 94 | 54 | 0.67 | 1.21(0.80,1.83) | 0.15 |
| | >20 km | 275 | 162 | 113 | | 1 | |
| **Family size** | <3 | 157 | 95 | 62 | 0.56 | 1.17(0.78,1.74) | 0.17 |
| | >3 | 266 | 151 | 115 | | 1 | |
| **Depression** | Yes | 237 | 132 | 105 | 17.6 | 2.37(1.60–3.52) | 0.001** |
| | No | 186 | 65 | 120 | | 1 | |
| **Intimate partner violence** | Yes | 144 | 90 | 54 | 30.4 | 3.18(2.09,4.83) | 0.02* |
| | No | 279 | 96 | 183 | | 1 | |
| **Social support** | Poor | 182 | 106 | 76 | 24.7 | 3.02(1.92, 4.75) | 0.01* |
| | Moderate | 89 | 36 | 53 | | 1.47 (0.85,2.54) | 0.29 |
| | Strong | 152 | 48 | 104 | | 1 | |
| **Duration of marriage** | < 3 year | 63 | 38 | 25 | 1.27 | 0.94(0.53, 1.67) | 0.11 |
| | 3–6 year | 148 | 99 | 49 | | 1.25(0.80, 1.94) | 0.14 |
| | >6year | 212 | 131 | 81 | 3.1205 | 1 | |

(1, reference category, * $p < 0.05$, **, $p < 0.01$), model fitness = 78%)

## Discussion

This community-based cross-sectional study design aimed to assess the prevalence of suicidal behavior and its associated factors among wives with polygamy marriage living in the Gedeo zone. This study found that 157 (37%) of the respondents had suicidal behavior, which was consistent with a study done in South Africa 39% [27], whereas this figure was higher than study conducted in Peru (22.6%) [28], Thailand (17.6%) [29], and Ethiopia (14%) [30]. This discrepancy could be explained by the difference in the character of study participants. Hence, the previous studies were conducted among pregnant, ante-natal and post-natal women. In addition, polygamous marriage is related to a lack of emotional support, economic dependency, and burden in the care of children, which could be the contributing risk factors to the higher suicidal behavior [31].

The current study has also identified variables such as non-formal educational status, more than two wives, current depression, intimate partner violence, and poor social supports which were associated suicidal behavior.

The respondents with no formal education/ illiterate had AOR 3.83;95% CI (2.44–6.02) more likely to develop suicidal behavior as compared to those with primary education, which was supported by the study conducted in the United States [32] and meta-analysis done in five low and middle income countries (India, South Africa, Nepal, Uganda, and Ethiopia) [33].

Illiteracy is related with lack of awareness, knowledge, and understanding regarding the women's rights, family, and marriage law of a country, which are vital to prevent intimate partner violence and abuse. Moreover, there is a tendency to be culturally more influenced in sharing their feelings frankly, opinions, and doubts happening in their marriage [34].

According to this study finding, being a wife of a husband with three and more other wives were AOR 2.29 95% CI (1.48, 3.53) more likely to experience suicidal behavior as compared to those with two wives, which was similar with studies done in Norway [35], Syria [36] and Turkey [37].

As the numbers of wives increase, the challenges of polygamy marriage such as social, emotional, economic crises become significant, which are common reasons for low self-esteem, helplessness, hopelessness, and thoughts of harming self [38].

Those respondents with a current history of depression had AOR 2.37 95% CI (1.60–3.52) more likely to show suicidal behavior, which was in- line with the previous studies conducted in Germany [39], South Africa [40], and meta-analysis done in Sub-Saharan African countries (Tanzania, Ethiopia, Nigeria, Uganda, Ghana and Burkina Faso) [41].

Depression is a common psychiatric disorder characterized by loss of interest, negative attitude for self, others, and the future, which all lead to the thoughts and actions of harming self [42].

According to this study result, respondents with a history of intimate partner violence had AOR 3.18 95% CI (2.09, 4.83) more likely to experience suicidal behavior. The finding in agreement with studies conducted in Thailand [29], Korea [43] and Sub- Sharan African counties (Angola, Burundi, Ethiopia, Uganda, Malawi, Mozambique, Zambia and Zimbabwe) [44].

Intimate partner violence has negative health consequence on the physical, emotional, psychological, and mental health of married women, which gradually cause serious psychiatry emergency conditions such as suicidal behavior [45].

The last predictor variable related to suicidal behavior was women's social support. Those with poor social support had AOR 3.02 95% CI (1.92, 4.75) more likely to develop suicidal behavior. Congruent findings were reported from the studies conducted in 23 European countries [46], South Africa [47] and Ethiopia [48]. Social support is a pillar for filling the gap of economic, social, and emotional crisis observed on women with polygamy marriage. Having no support results in helplessness, worthlessness, and low self-esteem, which are the main entry symptoms of depression and suicidal behavior [49].

## Strength and limitation of the study

This study tried to address very interesting issue regarding women's burden due to polygamy marriage. Moreover, the current study was conducted within traditional and cultural influenced community. Despite with the above strength, this study has also the following drawbacks. The cross-sectional nature of the study design constraint the cause effect relationship of the independent and outcome variables. In addition to this, some important variables such as husband's educational status, income, current wives pregnancy and health status were missed which might have correlation with polygamy marriage.

## Conclusion

This study found that more than one-third of wives with polygamy marriage had suicidal behavior. The most affected groups were those with no formal education, triple polygamy, current history of depression, intimate partner violence, and poor social support. Therefore, this harmful traditional and cultural practice should be avoided and reduced through the collaborative work of traditional and local community leaders, local and international women's rights association organizations, human rights associations, religious leaders, community mental health workers, and country ministry of health. In addition, empowering women with education and encouraging them in community affairs is an essential weapon to stop this type of marriage. Mental health professional should work in collaboration with stakeholders on screening and managing mental health conditions including suicidal behavior among communities with polygamy marriage.

## Acknowledgments

The author would like to thank study participants, Gedeo zone women counsel, and Dilla University for their invaluable support and recognition.

## Author Contributions

**Conceptualization:** Chalachew Kassaw, Seid Shumye.

**Data curation:** Seid Shumye.

**Formal analysis:** Seid Shumye.

**Investigation:** Chalachew Kassaw.

**Methodology:** Seid Shumye.

**Software:** Seid Shumye.

**Writing – original draft:** Chalachew Kassaw.

**Writing – review & editing:** Chalachew Kassaw.

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
