## [Decision Letter · Decision Letter 0]

20 Apr 2021

PONE-D-21-07591

The prevalence of suicidal behavior and its associated factors among wives with polygamy marriage living in Gedeo zone, Southern Ethiopia, 2020.

PLOS ONE

Dear Dr. Shumye,

Thank you for submitting your manuscript to PLOS ONE. After careful consideration, we feel that it has merit but does not fully meet PLOS ONE’s publication criteria as it currently stands. Therefore, we invite you to submit a revised version of the manuscript that addresses the points raised during the review process.

We look forward to receiving your revised manuscript.

Kind regards,

Shah Md Atiqul Haq

Academic Editor

PLOS ONE

Journal Requirements:

We note that you have indicated that data from this study are available upon request. PLOS only allows data to be available upon request if there are legal or ethical restrictions on sharing data publicly. For more information on unacceptable data access restrictions, please see http://journals.plos.org/plosone/s/data-availability#loc-unacceptable-data-access-restrictions.

2a) If there are ethical or legal restrictions on sharing a de-identified data set, please explain them in detail (e.g., data contain potentially sensitive information, data are owned by a third-party organization, etc.) and who has imposed them (e.g., an ethics committee). Please also provide contact information for a data access committee, ethics committee, or other institutional body to which data requests may be sent.

2b) If there are no restrictions, please upload the minimal anonymized data set necessary to replicate your study findings as either Supporting Information files or to a stable, public repository and provide us with the relevant URLs, DOIs, or accession numbers. For a list of acceptable repositories, please see http://journals.plos.org/plosone/s/data-availability#loc-recommended-repositories.

Your ethics statement should only appear in the Methods section of your manuscript. If your ethics statement is written in any section besides the Methods, please move it to the Methods section and delete it from any other section. Please ensure that your ethics statement is included in your manuscript, as the ethics statement entered into the online submission form will not be published alongside your manuscript.

Please include captions for your Supporting Information files at the end of your manuscript, and update any in-text citations to match accordingly. Please see our Supporting Information guidelines for more information: http://journals.plos.org/plosone/s/supporting-information.

Additional Editor Comments:

Dear authors,

I ask you to revise the document according to the reviewers' comments and suggestions. I understand that the document needs many improvements, including the English.

Please revise and resubmit for consideration for possible publication.

Good luck.

Reviewers' comments:

Reviewer's Responses to Questions

**Comments to the Author**

1. Is the manuscript technically sound, and do the data support the conclusions?

Reviewer #1: Yes

Reviewer #2: Yes

Reviewer #3: Partly

2. Has the statistical analysis been performed appropriately and rigorously? 

Reviewer #1: No

Reviewer #2: Yes

Reviewer #3: No

3. Have the authors made all data underlying the findings in their manuscript fully available?

Reviewer #1: Yes

Reviewer #2: Yes

Reviewer #3: Yes

4. Is the manuscript presented in an intelligible fashion and written in standard English?

Reviewer #1: Yes

Reviewer #2: Yes

Reviewer #3: No

5. Review Comments to the Author

Reviewer #1: The study aimed to assess the magnitude and associated factors of suicidal behavior among wives with polygamy marriage living in the Gedeo zone, Southern Ethiopia, 2020”

The point is interested and original, but the manuscript needs major revision

1- General comments

-The manuscript needs an English proofreading.

- results section lack adequate presentation. Insufficient descriptive analysis.

- conclusion isn’t consistent with the results.

- the researchers ignoring some confounding factors such as economic status, Having a pregnancy.

2- Specific comments

2-1 Introduction

- Some studies have measured suicides among women with certain characteristics in Ethiopia. Please mention them in the introduction, such as the study of (Belete and Misgan,2019), their study aimed to determine the prevalence of suicidal behavior in postpartum mothers.

2-2 Methods

- The sampling technique was not mentioned, line 2 mentioned only that systematic random sampling technique was used. It is advisable to mention how was this design achieved and how was the sample size calculated? Also, they should indicate inclusion and exclusion criteria and Response rate. A number of women who “completed the interview”, “refused participation”, “failed to complete the interview” “ number of cases excluded from analysis”.

- It is preferable to the Figure Sampling process of how women were selected systematically from the population. For example see (Belete and Misgan,2019).

2-3 results

- Why was the education variable limited to only two categories (non-formal & primary school) and neglected the other educational categories? Was the sample design limited to educated women with less than elementary education? I want more explanation.

- the researcher mentioned that “Descriptive statistics; frequency, percentage, mean and standard deviation were used to describe the socio-demographic characteristics of respondents” they haven’t continuous variables to calculate mean and SD as they mentioned.

- The researchers did not clearly define the dependent variable in the method section. It must be mentioned that it is a binary variable that takes two outcomes and so on…..

- the researchers should conduct hypothesis tests (chi square test) to examine whether Suicidal behavior differs across variable categories.

- Table (2): There is a fatal mistake, p-values must be mentioned for all variables and not only limited to the significant variables.

- in both Table (1) and (2),The researchers repeated needlessly the frequency distribution of the variables.

- interpretation of significance levels (*,**,***) is missing at table 2.

- Table titles should be rewritten. Title of Table (1) can be “Sociodemographic characteristics of women related to the presence or absence of suicidal behaviour”

- Researchers should have reviewed previous studies and extracted accurately the conceptual framework, the researchers have neglected main confounding variables that affect suicidal behavior and should have been taken into account such as "economic status of women", “Is she pregnant or postpartum mother at the time of the survey?”. The higher the economic status of the woman, the greater her chances of securing her future, enjoying more independence and avoiding suicide. Suicidal behaviour was found high among postpartum mothers and was associated with poor wealth economic status of the (Belete and Misgan,2019).

- When researchers excluded religion variable from the analysis ???

- the researchers didn’t assess the goodness of fit of the multiple logistic regression model. How do I know if their model fits the data? how well the model describes the observed data?. Without such an analysis, the inferences drawn from their model may be misleading or even totally incorrect.

2-4 Discussion:

Line 3 and line 6 repeat the same result ““This study found that 157 (37%) of the respondents had suicidal behavior”

- besides significant association with suicidal behaviour, The researchers should write (adjusted OR (AOR), 95%CI ).

- regarding “Depression is a common psychiatric disorder characterized by loss of interest, negative attitude for self, others, and the future, which all lead to the thoughts and actions of harming self”, they should add reference.

- the researchers didn’t clarify the limitations and strengths of their paper. Researchers should report limitations and potential controversies (if any) raised by the study. The sample design was preferable to contain a control group in order to measure a causal relationship between polygamy and suicidal behavior. also, the researchers have ignored the endogeneity problem that arisen from omitted variables, Wives with polygamy marriage are shaped by many factors such as the wife's health status, the husband's religion, and his educational level, these factors affect the dependent variable at the same time. Therefore, the independent variable “being Wives with polygamy marriage” is correlated with the random error which leads to biased estimates.

2-5 conclusion

- Although the researchers did not measure the effect of the woman's religion or that of her husband on suicide, they mentioned that “Almost all peoples living in this study area are protestant religion followers, and polygamy is against Christianity or the holy bible commandment. So, it would be better if leaders of the religion teach them the right and acceptable types of marriage in Christianity ". Recommendations should stem from the results of the study.

-

2-6 References

• I checked references and found that some references aren’t directly related to the contents of the manuscript, for example, reference#3 aimed to identify the number of synthetic cathinones mentioned in a range of psychonaut, NPS‐related, online sources; and describe the associated acute/long term clinical scenario and the related treatment/management. I need more explanation

Reviewer #2: 1. Summary of the research (summarizing the main research question, claims, and conclusions of the study. Also, providing the context for how this research fits within the existing literature.)

The study titled: “The prevalence of suicidal behavior and its associated factors among wives with polygamy marriage living in Gedeo zone, Southern Ethiopia, 2020” is a topical issue that is addressing a social concern. In the era of globalization and in the fourth industrial revolution epoch. Polygamy and suicidal behaviour is becoming increasingly common, especially in African countries. Accurate data regarding the polygamy scope in Africa is limited and its prevalence varies widely from country to country. Historically, factors such as demographic and socioeconomic transitions have been identified that may appear to perpetuate polygamy, hence suicidal behaviour follows when the marriage is no longer palatable, especially among younger women. Thus, this study has brought out its relevance by carrying out a study using Ethiopia as a case study. Ethiopia is faced with growing prevalence of polygamy marriages as well as suicide behaviour among women in polygamous marriages (I have inserted some Readings in the main work pop up comments).

Manuscript’s strengths: The strength of the manuscript was a study carried out as a cross-sectional survey and also one of the few studies on this topical issue above discussed.

Manuscript’s weaknesses: The manuscript weaknesses was the author had failed to include prevalences of the topical issues and the introduction and methodology is lagging behind (See all my comments on the pop up comment chat on the main body of the paper).

2. Major issues

Major issues were inserted in the body of work (see the yellow pop up comment chat).

Minor issues

Minor issues were inserted in the body of work (see the yellow pop up comment chat).

3. Other points: Get more readings from African countries on polygamy marriage and suicidal behaviour.

4. Overall recommendation: I recommend this study titled: “The prevalence of suicidal behavior and its associated factors among wives with polygamy marriage living in Gedeo zone, Southern Ethiopia, 2020” for publication with major revision (See all comments and suggested readings that will help the work).

Reviewer #3: Thank you for giving the opportunity to review this interesting article. Hope my comments would be helpful to increase the quality of your work.

Abstract: Avoid the repetition in the abstract. Factors such as illiteracy, being a wife of a husband with three and more other wives, current history of depression, intimate partner violence, etc have been mentioned in both results and conclusion sections.

Introduction:

• There is a clear need that this paper needs to be proofread, as there are some grammatical and typography errors throughout the document.

• “Developed countries were launched human and legal rights.” can you mention few countries, this has to be specific.

• Does the literature gather any numerical evidence on the cases on polygamy in Ethiopia? How would you justify that there is a high prevalence of these practises?

Study participants: How many respondents have you selected for the study and how many respondents have you excluded? There should be some information on how you have chosen the respondents. Have you used any sampling frame?

Study Design, Setting and Period:

“wives with polygamy marriage from Nov – Dec 2020” this sentence is not clear; you could add that this study interviewed the women who have been married at the time month of (interview date-November)

Data collection and instruments:

• It would be better to include a table on scores as mentioned in the section, instrument. Results seem rather dramatic.

• Have you been referred to the “questionnaire” ? or a person who asked the questions, as the author used the word “questioner”, this is not clear. This could be seen throughout the paper.

• Variable categories seem restricted that could not see any variations within. If you have collected more information, it would be good to add these. For instance, for age groups, educational status and employment categories. How would you define primary education here? Is it for those who completed up to grade 5 at school? be specific here.

Psychosocial results of respondents

• A summary table may be good to pull the results here, can you mention about respondents who may had depression, intimate partner violence, suicidal behaviour or two of these according to the scores?

• Methods section need to be redesigned, not very clear on what are the outcome variable, independent variable categories and suicide behaviour has been used as a binary variable coded in to two categories.

Factors associated with suicidal behaviour

• Use the standard format when reporting the tables. Check the format of refereed journal articles. There are visible errors in reporting numbers, lack of consistency in gaps and after AOR, add the heading for confidence intervals.

• Prior to the logistic regression have you done a chi-square test to see any association? This is not clear here. Have you found any insignificant variables here?

• Include the significance threshold at the end of the table which are denoted by stars.

Discussion

• This sentence “This study found that 157 (37 %) of the respondents had suicidal behaviour”, has been repeated twice, avoid repetition.

• There is a need of including citation in several places in the discussion, for instance when referring to the cultural difference and contextual definition of having more than one wife.

• Define what is meant by (2.44-6.02), proper way would be AOR 3.83;95% CI 2.44-6.02), repeat it for other places. Mention that 1 has been used as a reference category.

• Why you did not discuss on duration of marriage influence on suicidal behaviour? Have you found any literature in the context of Africa? Can you justify this finding?

6. PLOS authors have the option to publish the peer review history of their article (what does this mean?). If published, this will include your full peer review and any attached files.

Reviewer #1: **Yes: **Suzan Abdel-Rahman

Reviewer #2: **Yes: **Monica Ewomazino Akokuwebe, PhD

Reviewer #3: No

---

## [Author Response · Author response to Decision Letter 0]

18 Jun 2021

Point by point response letter

Dear editor and reviewers,

 We would like to acknowledge for your detailed and professionally sounded review of our manuscript. We have accessed and found constructive and very important comments to improve the quality of the manuscript. We have taken time to revise the manuscript and all the comments raised are well addressed. We also did a great revision of the manuscript to improve the English language quality.

 Please find the revised version of the manuscript with a clean and track change form together with this point by point response letter. In this response letter, our reply is found next to the comments raised for each concern. If addition revision is needed, we are ready to modify it.

Thank you

Regards 

Seid Shumye

Corresponding author

Reviewer reports

Reviewer #1: The study aimed to assess the magnitude and associated factors of suicidal behavior among wives with polygamy marriage living in the Gedeo zone, Southern Ethiopia, 2020”

The point is interested and original, but the manuscript needs major revision

1. Reply: Thank you very much. We have accepted the comment and modified accordingly. 

General comments

-The manuscript needs English proofreading.

 2. Reply: Thank you. We did a great revision of the manuscript to improve the English language quality.

- results section lack adequate presentation. Insufficient descriptive analysis.

3. Reply: we accept the comment and described the findings accordingly.

- conclusion isn’t consistent with the results.

4. Reply: Thank you very much. We accept the comment and revised properly.

- the researchers ignoring some confounding factors such as economic status, Having a pregnancy.

5. Replay: Thank you very much for your suggestion. We accept the comment and stated on the limitation section

2- Specific comments

2-1 Introduction

- Some studies have measured suicides among women with certain characteristics in Ethiopia. Please mention them in the introduction, such as the study of (Belete and Misgan,2019), their study aimed to determine the prevalence of suicidal behavior in postpartum mothers.

6. Reply: Thank you. We accept the comment and incorporated studies which have been done on this issue.

2-2 Methods

- The sampling technique was not mentioned, line 2 mentioned only that systematic random sampling technique was used. It is advisable to mention how was this design achieved and how was the sample size calculated? Also, they should indicate inclusion and exclusion criteria and Response rate. A number of women who “completed the interview”, “refused participation”, “failed to complete the interview” “number of cases excluded from analysis”.

7. Reply: Thank you very much. We accept the comment and tried to clearly mention the flow of the sampling technique as well as the sample size calculation using the standard formula.

- It is preferable to the Figure Sampling process of how women were selected systematically from the population. For example see (Belete and Misgan,2019).

8. Reply: Thank you for your suggestion. We have seen the sampling process of Belete’s study. Since our study was focused only on the community, we clearly stated the process on the revised manuscript. 

2-3 results

- Why was the education variable limited to only two categories (non-formal & primary school) and neglected the other educational categories? Was the sample design limited to educated women with less than elementary education? I want more explanation.

9. Reply: Thank you very much. We accept the comment and appreciated your concern. During our data collection, we incorporated different education categories in the questionnaire but in the result section we included the education categories based on our finding from the questionnaire. 

- the researcher mentioned that “Descriptive statistics; frequency, percentage, mean and standard deviation were used to describe the socio-demographic characteristics of respondents” they haven’t continuous variables to calculate mean and SD as they mentioned.

10. Reply: Thank you. We accept the comment and incorporate age as a continuous variable to calculate mean and SD.

- The researchers did not clearly define the dependent variable in the method section. It must be mentioned that it is a binary variable that takes two outcomes and so on…..

11. Replay: Thank you very much. We accept the comment and included the outcome variable (presence or absence of suicidal behavior.

- the researchers should conduct hypothesis tests (chi square test) to examine whether Suicidal behavior differs across variable categories.

12. Replay: Thank you very much. We accept the comment. Yes, we conducted the chi square test to see the suicidal behavior across the variables and to identify the candidate variables to multivariate analysis.

- Table (2): There is a fatal mistake, p-values must be mentioned for all variables and not only limited to the significant variables.

13. Reply: Thank you very much. We accept the comment and incorporate the p-values for all variables.

- in both Table (1) and (2),The researchers repeated needlessly the frequency distribution of the variables.

14. Reply: Thank you. 

- interpretation of significance levels (*,**,***) is missing at table 2.

15. Reply: Thank you very much. We accept the comment and incorporated accordingly.

- Table titles should be rewritten. Title of Table (1) can be “Sociodemographic characteristics of women related to the presence or absence of suicidal behaviour”

16. Reply: Thank you very much. We accept the comment and modified accordingly.

- Researchers should have reviewed previous studies and extracted accurately the conceptual framework, the researchers have neglected main confounding variables that affect suicidal behavior and should have been taken into account such as "economic status of women", “Is she pregnant or postpartum mother at the time of the survey?”. The higher the economic status of the woman, the greater her chances of securing her future, enjoying more independence and avoiding suicide. Suicidal behaviour was found high among postpartum mothers and was associated with poor wealth economic status of the (Belete and Misgan,2019).

17. Reply: Thank you very much for your suggestion. We accept the comment and stated on the limitation section

- When researchers excluded religion variable from the analysis???

18. Reply: Thank you very much. We accept the comment. Religion variable was excluded during bivariate analysis.

- the researchers didn’t assess the goodness of fit of the multiple logistic regression model. How do I know if their model fits the data? how well the model describes the observed data?. Without such an analysis, the inferences drawn from their model may be misleading or even totally incorrect.

19. Reply: Thank you very much. We accept the comment and incorporated the model fitness accordingly.

2-4 Discussion:

Line 3 and line 6 repeat the same result ““This study found that 157 (37%) of the respondents had suicidal 

behavior”

20. Replay: Thank you. We have omitted the unnecessary repetition.

- besides significant association with suicidal behaviour, The researchers should write (adjusted OR (AOR), 95%CI ).

21. Replay: thank you very much. We accept the comment and incorporated accordingly.

- regarding “Depression is a common psychiatric disorder characterized by loss of interest, negative attitude for self, others, and the future, which all lead to the thoughts and actions of harming self”, they should add reference.

22. Replay: Thank you very much. We accept the comment and put the reference.

- the researchers didn’t clarify the limitations and strengths of their paper. Researchers should report limitations and potential controversies (if any) raised by the study. The sample design was preferable to contain a control group in order to measure a causal relationship between polygamy and suicidal behavior. also, the researchers have ignored the endogeneity problem that arisen from omitted variables, Wives with polygamy marriage are shaped by many factors such as the wife's health status, the husband's religion, and his educational level, these factors affect the dependent variable at the same time. Therefore, the independent variable “being Wives with polygamy marriage” is correlated with the random error which leads to biased estimates.

23. Replay: Thank you very much. We accept the comment and incorporated your concerns in the limitation section. 

2-5 conclusion

- Although the researchers did not measure the effect of the woman's religion or that of her husband on suicide, they mentioned that “Almost all peoples living in this study area are protestant religion followers, and polygamy is against Christianity or the holy bible commandment. So, it would be better if leaders of the religion teach them the right and acceptable types of marriage in Christianity ". Recommendations should stem from the results of the study.

24. Reply: Thank you very much. We accept the comment and put the recommendation based on our results. 

-

2-6 References

• I checked references and found that some references aren’t directly related to the contents of the manuscript, for example, reference#3 aimed to identify the number of synthetic cathinones mentioned in a range of psychonaut, NPS‐related, online sources; and describe the associated acute/long term clinical scenario and the related treatment/management. I need more explanation

25. Replay: Thank you very much. We accept the comment. We have checked all references again and made the corrections properly.

Reviewer #2: 1. Summary of the research (summarizing the main research question, claims, and conclusions of the study. Also, providing the context for how this research fits within the existing literature.)

The study titled: “The prevalence of suicidal behavior and its associated factors among wives with polygamy marriage living in Gedeo zone, Southern Ethiopia, 2020” is a topical issue that is addressing a social concern. In the era of globalization and in the fourth industrial revolution epoch. Polygamy and suicidal behaviour is becoming increasingly common, especially in African countries. Accurate data regarding the polygamy scope in Africa is limited and its prevalence varies widely from country to country. Historically, factors such as demographic and socioeconomic transitions have been identified that may appear to perpetuate polygamy, hence suicidal behaviour follows when the marriage is no longer palatable, especially among younger women. Thus, this study has brought out its relevance by carrying out a study using Ethiopia as a case study. Ethiopia is faced with growing prevalence of polygamy marriages as well as suicide behaviour among women in polygamous marriages (I have inserted some Readings in the main work pop up comments).

Manuscript’s strengths: The strength of the manuscript was a study carried out as a cross-sectional survey and also one of the few studies on this topical issue above discussed.

Manuscript’s weaknesses: The manuscript weaknesses was the author had failed to include prevalences of the topical issues and the introduction and methodology is lagging behind (See all my comments on the pop up comment chat on the main body of the paper).

2. Major issues

Major issues were inserted in the body of work (see the yellow pop up comment chat).

1. Reply: Thank you very much for your constructive comments and suggestions. We accept the comments and suggestions and revised the manuscript accordingly. 

Minor issues

Minor issues were inserted in the body of work (see the yellow pop up comment chat).

2. Reply: Thank you very much. We accept the comments and modified accordingly.

Other points: Get more readings from African countries on polygamy marriage and suicidal behaviour.

3. Reply: Thank you very much. We have incorporated studies which have done in Africa in the revised manuscript

 Overall recommendation: I recommend this study titled: “The prevalence of suicidal behavior and its associated factors among wives with polygamy marriage living in Gedeo zone, Southern Ethiopia, 2020” for publication with major revision (See all comments and suggested readings that will help the work).

Reviewer #3: Thank you for giving the opportunity to review this interesting article. Hope my comments would be helpful to increase the quality of your work.

Abstract: Avoid the repetition in the abstract. Factors such as illiteracy, being a wife of a husband with three and more other wives, current history of depression, intimate partner violence, etc have been mentioned in both results and conclusion sections.

1. Reply: Thank you very much. We accept the comment and avoided the unnecessary repetition. 

Introduction:

• There is a clear need that this paper needs to be proofread, as there are some grammatical and typography errors throughout the document.

2. Reply: Thank you very much. We have revised the grammatical/ language editing by English language experts in all components of the manuscript.

• “Developed countries were launched human and legal rights.” can you mention few countries, this has to be specific.

3. Reply: yes, we can mention few developed countries which launched human and legal rights to condemned polygamous marriage. Such as Canada, USA, and Australia..

• Does the literature gather any numerical evidence on the cases on polygamy in Ethiopia? How would you justify that there is a high prevalence of these practises?

4. Reply: Thank you very much. We have found study which has done in Ethiopia. In Ethiopia, polygamy cultural practices accounts 14 %.

Study participants: How many respondents have you selected for the study and how many respondents have you excluded? There should be some information on how you have chosen the respondents. Have you used any sampling frame?

5. Reply: Thank you very much. We accept the comment and incorporated the total samples for the current study. We have clearly stated how we chose the respondents in the revised manuscript.

We used the respondent’s household as a sampling frame by assigned marks on their doors.

Study Design, Setting and Period:

“wives with polygamy marriage from Nov – Dec 2020” this sentence is not clear; you could add that this study interviewed the women who have been married at the time month of (interview date-November)

6. Reply: Thank you very much. We accept the comment and modified accordingly.

Data collection and instruments:

• It would be better to include a table on scores as mentioned in the section, instrument. Results seem rather dramatic.

7. Reply: Thank you for your suggestion to include item scores. We did it for the outcome variable (suicidal behavior) accordingly.

• Have you been referred to the “questionnaire” ? or a person who asked the questions, as the author used the word “questioner”, this is not clear. This could be seen throughout the paper.

8. Reply: Thank you for your concern. We modified (replaced) the term “questioner “by “questionnaire” throughout the document. 

• Variable categories seem restricted that could not see any variations within. If you have collected more information, it would be good to add these. For instance, for age groups, educational status and employment categories. How would you define primary education here? Is it for those who completed up to grade 5 at school? be specific here.

9. Reply: Thank you very much. We would like you acknowledge your concern. Of, course we have included extended categories in the questionnaire but after we collected the data we categorized based on our findings and used different literatures. 

Psychosocial results of respondents

• A summary table may be good to pull the results here, can you mention about respondents who may had depression, intimate partner violence, suicidal behaviour or two of these according to the scores?

10. Reply: Thank you. It is very nice comment and we summarized depression and intimate partner violence results using the bar graph (figure). For the first objective; prevalence of suicidal behavior among wives with polygamous marriage; the results were shown using table according to the scores.

• Methods section need to be redesigned, not very clear on what are the outcome variable, independent variable categories and suicide behaviour has been used as a binary variable coded in to two categories.

11. Reply: Thank you very much. We accept the comment and made a clear statement in the revised manuscript. We put the study variable as sub section like the outcome variable (presence or absence of suicide) and independent variable categories in the revised manuscript. 

Factors associated with suicidal behaviour

• Use the standard format when reporting the tables. Check the format of refereed journal articles. There are visible errors in reporting numbers, lack of consistency in gaps and after AOR, add the heading for confidence intervals.

12. Reply: Thank you. We accept the comment and modified the table using the standard format

• Prior to the logistic regression have you done a chi-square test to see any association? This is not clear here. Have you found any insignificant variables here?

13. Reply: we appreciate your comment. Yes, we did a chi-square test to see the association between the independent and dependent variables and to the select the candidate variables for the logistic regression analysis. We put a chi-square test results in the revised manuscript.

• Include the significance threshold at the end of the table which are denoted by stars.

14. Reply: Thank you. We accept the comment and incorporated the significant threshold denoted by stars at the bottom of the table.

Discussion

• This sentence “This study found that 157 (37 %) of the respondents had suicidal behaviour”, has been repeated twice, avoid repetition.

15. Reply: Thank you. We accept the comment and modified accordingly.

• There is a need of including citation in several places in the discussion, for instance when referring to the cultural difference and contextual definition of having more than one wife.

16. Reply: Thank you very much. We accept the comment and put the references accordingly.

• Define what is meant by (2.44-6.02), proper way would be AOR 3.83;95% CI 2.44-6.02), repeat it for other places. Mention that 1 has been used as a reference category.

17. Reply: it is very nice comment. We accept the comment and modified accordingly.

• Why you did not discuss on duration of marriage influence on suicidal behaviour? Have you found any literature in the context of Africa? Can you justify this finding?

18. Reply: Thank you very much. We acknowledge your concern. The reason why we didn’t discuss duration of marriage; it wasn’t the significant variable during the logistic regression analysis output.

---

## [Decision Letter · Decision Letter 1]

11 Jul 2021

PONE-D-21-07591R1

The prevalence of suicidal behavior and its associated factors among wives with polygamy marriage living in Gedeo zone, southern Ethiopia, 2020.

PLOS ONE

Dear Dr. Seid Shumye,

Thank you for submitting your manuscript to PLOS ONE. After careful consideration, we feel that it has merit but does not fully meet PLOS ONE’s publication criteria as it currently stands. Therefore, we invite you to submit a revised version of the manuscript that addresses the points raised during the review process.

We look forward to receiving your revised manuscript.

Kind regards,

Shah Md Atiqul Haq

Academic Editor

PLOS ONE

Additional Editor Comments (if provided):

Dear Authors,

Based on the advice I suggest to revise the paper by following the reviewers' comments and suggestions.

Please resubmit the revised version.

Best regards,

Reviewers' comments:

Reviewer's Responses to Questions

**Comments to the Author**

1. If the authors have adequately addressed your comments raised in a previous round of review and you feel that this manuscript is now acceptable for publication, you may indicate that here to bypass the “Comments to the Author” section, enter your conflict of interest statement in the “Confidential to Editor” section, and submit your "Accept" recommendation.

Reviewer #1: (No Response)

Reviewer #3: (No Response)

2. Is the manuscript technically sound, and do the data support the conclusions?

Reviewer #1: Yes

Reviewer #3: Yes

3. Has the statistical analysis been performed appropriately and rigorously? 

Reviewer #1: No

Reviewer #3: Yes

4. Have the authors made all data underlying the findings in their manuscript fully available?

Reviewer #1: No

Reviewer #3: Yes

5. Is the manuscript presented in an intelligible fashion and written in standard English?

Reviewer #1: Yes

Reviewer #3: Yes

6. Review Comments to the Author

Reviewer #1: I am not satisfied with the responses.Statistical analysis is unreliable.

he paper lacks sufficient quality for publication, I evaluated it as a whole, but I was not convinced of it,I am not satisfied with the responses the researchers did not make the modifications with the required quality.

Reviewer #3: There is a problem with keeping spaces consistent throughout.

The fertility rate is high in this area as compared to other parts of the country. This needs to be cited.

Add few sentences on systematic random sampling technique. It seems quite arbitrary.

Results

My confusion still remains as to why your age, husband's work, and educational level sections are so limited. Adding more categories would be helpful.

It would be beneficial to add a brief paragraph to describe the outcome variables and other covariates (see the standard journal article format). Additionally, you were unable to refer to 1 since it is used as a reference category.

Despite your acknowledgment in your comments, I have not been able to understand when you used the Chi-square test. It is of major concern that some variables, such as religion, were left out of the AOR model and that some variables, such as depression, and domestic violence, were absent from the first table. This needs to be clearly specified. Additionally, you could replace the first table with chi-square results and 95%CI intervals, since it makes no sense to repeat percentages in both tables.

Discussion

Why haven't you compared your results with previous studies in Africa in the Discussion?

Is that due to the lack of studies? As well, I suggest reviewing a standard discussion of a journal article that explains the results critically rather than simply reporting them.

The study should also be mentioned with its strengths and limitations -which is a glaring omission.

7. PLOS authors have the option to publish the peer review history of their article (what does this mean?). If published, this will include your full peer review and any attached files.

Reviewer #1: **Yes: **suzan Abdel-Rahman mohamed

Reviewer #3: **Yes: **Gayathri Abeywickrama

---

## [Author Response · Author response to Decision Letter 1]

4 Aug 2021

Additional Editor Comments (if provided):

Dear Authors,

Based on the advice I suggest to revise the paper by following the reviewers' comments and suggestions.

Please resubmit the revised version.

Best regards,

Reviewers' comments:

Reviewer's Responses to Questions

Comments to the Author

1. If the authors have adequately addressed your comments raised in a previous round of review and you feel that this manuscript is now acceptable for publication, you may indicate that here to bypass the “Comments to the Author” section, enter your conflict of interest statement in the “Confidential to Editor” section, and submit your "Accept" recommendation.

Reviewer #1: (No Response)

Reviewer #3: (No Response)

2. Is the manuscript technically sound, and do the data support the conclusions?

Reviewer #1: Yes

Reviewer #3: Yes

3. Has the statistical analysis been performed appropriately and rigorously?

Reviewer #1: No

Reviewer #3: Yes

4. Have the authors made all data underlying the findings in their manuscript fully available?

Reviewer #1: No

Reviewer #3: Yes

5. Is the manuscript presented in an intelligible fashion and written in standard English?

Reviewer #1: Yes

Reviewer #3: Yes

6. Review Comments to the Author

1. Reviewer #1: I am not satisfied with the responses. Statistical analysis is unreliable.

The paper lacks sufficient quality for publication, I evaluated it as a whole, but I was not convinced of it, I am not satisfied with the responses the researchers did not make the modifications with the required quality.

Author’s response: Dear reviewer, we authors want to ask excuse for your feeling of being not satisfied with our response. 

We tried to revise again your comments and incorporate responses. 

Note: We authors are ready to correct each specific question you want to raise for clarification and elaboration again. We authors are also ready to submit the data if you have such significant concern. 

2. Reviewer #3: There is a problem with keeping spaces consistent throughout.

Author’s response: Thank you dear reviewer. We accept the comment and corrected accordingly.

The fertility rate is high in this area as compared to other parts of the country. This needs to be cited.

Author’s response: Thank you dear reviewer. 

The reference you asked was cited properly and written as follows 

The zone has a population of 850, 534 and 500,000 of them were females. The fertility rate is higher in this area as compared to other parts of the country (22). 

3. Add few sentences on systematic random sampling technique. It seems quite arbitrary.

 Author’s response: Dear reviewer, we included few sentences and re- written as follows 

Sampling Technique

This study used a systematic random sampling technique. To get the sampling frame of the sample, we gave coding for each number households practicing polygamy marriage. There were 1692 households who practiced polygamy and living in the Gedeo zone. To calculate Kth interval, we used the formula (N/n), where N= Total number of population and n= required sample. Kth interval= (1692/423=4th) and include all samples by counting every 4th interval.

Results

4. My confusion still remains as to why your age, husband's work, and educational level sections are so limited. Adding more categories would be helpful.

Author’s response: Dear reviewer, our participant’s data was in to two for Age, Husband work and educational classification. 

This is based on the finding of our data. 

5. It would be beneficial to add a brief paragraph to describe the outcome variables and other covariates (see the standard journal article format). 

Author’s response: Dear reviewer, it was corrected as follows 

Study variables 

Dependent/out-come variable 

Suicidal behaviour (Yes/No) 

Independent variables 

Socio-demographic variables: Age, educational status, employment, residence, husband work, wife’s house

Marriage and psycho-social variables: religion, and duration of marriage, number of wives at a time, distance of the two wife’s house, family size, social support, and duration of marriage. 

6. Additionally, you were unable to refer to 1 since it is used as a reference category.

Author’s response: we were rewritten on the bottom of the table as follows 

(1, reference category, * p< 0.05, **, p<0.01), model fitness = 78%) 

7. Despite your acknowledgment in your comments, I have not been able to understand when you used the Chi-square test.

Authors response: Dear reviewer one of the reviewers suggested to include chi-square 

8. It is of major concern that some variables, such as religion, were left out of the AOR model.

Authors response: Dear reviewer, we included and corrected as follow 

Religion Protestant 329(77.7% 172 157 1 

 Orthodox 64(15.1%) 29 35 1.32(0.77- 2.26) 

 Muslim 30(7.09) 14 16 1.252(0.59- 2.64) 

 9. And that some variables, such as depression, and domestic violence, were absent from the first table.

 Author’s response: Thank you, Dear reviewer; it was there in table 2

Depression Yes 237(56.0%) 132 105 17.6 .000027 

 No 186(43.9%) 65 120 

Intimate partner violence Yes 144(43.0%) 90 54 30.4 0.00001 

 No 279(65.9%) 96 183 

10. This needs to be clearly specified. Additionally, you could replace the first table with chi-square results and 95%CI intervals, since it makes no sense to repeat percentages in both tables.

Author’s response: Dear reviewer, we merged both table 1 and table 2 by including the Chi-square result 

Variables Category variables Suicidal behavior 

 X2 Adjusted odds ratio (AOR), 95% CI P-value 

 Yes No 

 Age Below 30 135 94 41 3.1205 1.48(0.96, 2.29) 0.43

 Above 30 288 175 113 1 

Educational status Non formal(illiterate) 309 211 98 34.7904 3.83 (2.44-6.02) 0.001*

 Primary school 

(grade 1-8) 114 41 73 1 

Residence Urban 42 19 23 3.22 0.53(0.28,1.01) 0.13

 Rural 381 232 149 1 

Religion Protestant 329(77.7% 172 157 1.6788 1 

 Orthodox 64(15.1%) 29 35 1.32(0.77- 2.26) 

 Muslim 30(7.09) 14 16 1.252(0.59- 2.64) 

Husbands work Farmer 313 140 173 0.83 1 

 Merchant 110 43 67 0.79(0.51,1.24) 0.24

Wives work House wife 250 134 116 5.71 1 

 Farmer 89 39 50 0.68(0.41,1.10) 0.10

 Merchant 84 52 32 1.41(0.85, 2.33) 0.37

Number of wives at a time Two 300 132 168 13.4 1 

 Three 123 79 44 2.29(1.48,3.53) 0.01*

Distance of the two wives house < 20 km 148 94 54 0.67 1.21(0.80,1.83) 0.15

 >20 km 275 162 113 1 

Family size <3 157 95 62 0.56 1.17(0.78,1.74) 0.17

 >3 266 151 115 1 

Depression Yes 237 132 105 17.6 2.37(1.60-3.52) 0.001**

 No 186 65 120 1 

Intimate partner violence Yes 144 90 54 30.4 3.18(2.09,4.83) 0.02*

 No 279 96 183 1 

Social support Poor 182 106 76 24.7 3.02(1.92, 4.75) 0.01*

 Moderate 89 36 53 1.47 (0.85,2.54) 0.29

 Strong 152 48 104 1 

Duration of marriage < 3 year 63 38 25 1.27 0.94(0.53, 1.67) 0.11

 3-6 year 148 99 49 1.25(0.80, 1.94) 0.14

 >6year 212 131 81 3.1205 1 

Discussion

11. Why haven't you compared your results with previous studies in Africa in the Discussion?

Is that due to the lack of studies? As well, I suggest reviewing a standard discussion of a journal article that explains the results critically rather than simply reporting them. 

Author’s response: Thank you very much for great suggestions. 

We included African studies such as South Africa, Uganda and Ghana 

The study should also be mentioned with its strengths and limitations -which is a glaring omission.

Author’s response: Thank you very much. We accept the comment and added the strength and limitation of our study in the revised manuscript.

7. PLOS authors have the option to publish the peer review history of their article (what does this mean?). If published, this will include your full peer review and any attached files.

Do you want your identity to be public for this peer review? For information about this choice, including consent withdrawal, please see our Privacy Policy.

Reviewer #1: Yes: suzan Abdel-Rahman mohamed

Reviewer #3: Yes: Gayathri Abeywickrama

---

## [Decision Letter · Decision Letter 2]

12 Oct 2021

The prevalence of suicidal behavior and its associated factors among wives with polygamy marriage living in Gedeo zone, southern Ethiopia, 2020.

PONE-D-21-07591R2

Dear Dr. Seid Shumye,

We’re pleased to inform you that your manuscript has been judged scientifically suitable for publication and will be formally accepted for publication once it meets all outstanding technical requirements.

Kind regards,

Shah Md Atiqul Haq

Academic Editor

PLOS ONE

Additional Editor Comments (optional):

Dear authors,

Thank you for addressing the reviewers' comments and suggestions.

Before final acceptance of the paper, I would ask you check English of the paper by a native.

Best wishes

Reviewers' comments:

Reviewer's Responses to Questions

**Comments to the Author**

1. If the authors have adequately addressed your comments raised in a previous round of review and you feel that this manuscript is now acceptable for publication, you may indicate that here to bypass the “Comments to the Author” section, enter your conflict of interest statement in the “Confidential to Editor” section, and submit your "Accept" recommendation.

Reviewer #3: All comments have been addressed

2. Is the manuscript technically sound, and do the data support the conclusions?

Reviewer #3: Yes

3. Has the statistical analysis been performed appropriately and rigorously? 

Reviewer #3: Yes

4. Have the authors made all data underlying the findings in their manuscript fully available?

Reviewer #3: Yes

5. Is the manuscript presented in an intelligible fashion and written in standard English?

Reviewer #3: Yes

6. Review Comments to the Author

Reviewer #3: Authors tried to address the comments raised by me. It has been improved compared to the first draft. Good luck with your work.

7. PLOS authors have the option to publish the peer review history of their article (what does this mean?). If published, this will include your full peer review and any attached files.

Reviewer #3: No

---

## [Editor Report · Acceptance letter]

15 Oct 2021

PONE-D-21-07591R2 

The prevalence of suicidal behavior and its associated factors among wives with polygamy marriage living in Gedeo zone, southern Ethiopia, 2020. 

Dear Dr. Shumye:

I'm pleased to inform you that your manuscript has been deemed suitable for publication in PLOS ONE. Congratulations! Your manuscript is now with our production department. 

Kind regards, 

on behalf of

Dr. Shah Md Atiqul Haq 

Academic Editor

PLOS ONE